# S-Methylmethionine Effectively Alleviates Stress in Szarvasi-1 Energy Grass by Reducing Root-to-Shoot Cadmium Translocation

**DOI:** 10.3390/plants11212979

**Published:** 2022-11-04

**Authors:** Deepali Rana, Vitor Arcoverde Cerveira Sterner, Aravinda Kumar Potluri, Zoltán May, Brigitta Müller, Ádám Solti, Szabolcs Rudnóy, Gyula Sipos, Csaba Gyuricza, Ferenc Fodor

**Affiliations:** 1Department of Plant Physiology and Molecular Plant Biology, ELTE Eötvös Loránd University, Pázmány Péter Lane 1/c, 1117 Budapest, Hungary; 2Doctoral School of Environmental Sciences, ELTE Eötvös Loránd University, Pázmány Péter Lane 1/a, 1117 Budapest, Hungary; 3Doctoral School of Biological Sciences, ELTE Eötvös Loránd University, Pázmány Péter Lane 1/c, 1117 Budapest, Hungary; 4Institute of Materials and Environmental Chemistry, Research Centre for Natural Sciences, Eötvös Loránd Research Network, Magyar Tudósok Blvd. 2, 1117 Budapest, Hungary; 5Agricultural Research and Development Institute, Szabadság Street 30, 5540 Szarvas, Hungary; 6Institute of Agronomy, Hungarian University of Agriculture and Life Sciences, Páter Károly Street 1, 2100 Gödöllő, Hungary

**Keywords:** biomass plant, element composition, *Elymus elongatus*, heavy metal stress, metal transport, phytoremediation, priming, *Thinopyrum obtusiflorum*

## Abstract

S-methylmethionine (SMM) is a universal metabolite of higher plants derived from L-methionine that has an approved priming effect under different types of abiotic and biotic stresses. Szarvasi-1 energy grass (*Elymus elongatus* subsp. *ponticus* cv. Szarvasi-1) is a biomass plant increasingly applied in phytoremediation to stabilize or extract heavy metals. In this study, Szarvasi-1 was grown in a nutrient solution. As a priming agent, SMM was applied in 0.02, 0.05 and 0.1 mM concentrations prior to 0.01 mM Cd addition. The growth and physiological parameters, as well as the accumulation pattern of Cd and essential mineral nutrients, were investigated. Cd exposure decreased the root and shoot growth, chlorophyll concentration, stomatal conductance, photosystem II function and increased the carotenoid content. Except for stomatal conductance, SMM priming had a positive effect on these parameters compared to Cd treatment without priming. In addition, it decreased the translocation and accumulation of Cd. Cd treatment decreased K, Mg, Mn, Zn and P in the roots, and K, S, Cu and Zn in the shoots compared to the untreated control. SMM priming changed the pattern of nutrient uptake, of which Fe showed characteristic accumulation in the roots in response to increasing SMM concentrations. We have concluded that SMM priming exerts a positive effect on Cd-stressed Szarvasi-1 plants, which retained their physiological performance and growth. This ameliorative effect is suggested to be based on, at least partly, the lower root-to-shoot Cd translocation by the upregulated Fe uptake and transport.

## 1. Introduction

Multiple anthropogenic activities, such as using wastewater for irrigation, mining, the application of sewage sludge and phosphate fertilizers in agriculture, transportation and industrial activities, are responsible for the accumulation of heavy metals in soil [1,2,3]. Among heavy metals, cadmium is a non-essential element that is extremely poisonous, even at very low concentrations, to plants and animals [4]. After entering into plants, Cd is responsible for various physiological and biochemical disorders [5]. Moreover, Cd tends to accumulate in edible aerial parts, such as the grains and leaves of plants grown in soil contaminated, with a moderate amount of Cd below the toxicity threshold [6,7]. Due to the continuous contamination of agricultural soils by Cd, this environmental pollutant has become one of the most severe and widespread agricultural and environmental challenges worldwide [1,8,9].

The mobility of Cd in agricultural soil is fundamentally influenced by the soil pH. Under acidic conditions, Cd is generally soluble in water, but its solubility is limited by the formation of CdCO_3_ and CdS or the adsorption of Cd onto the solid phase of soil in neutral to alkaline soil [10,11,12,13]. The cation exchange capacity of soil also influences Cd mobility in soil [14]. Generally, in unpolluted soils Cd contamination has a geogenic origin, thus, Cd concentration is generally less than 0.5 mg kg^−1^ dry soil. Due to the different kinds of anthropogenic activities, this basic level of Cd might reach up to 3.0 mg kg^−1^, or even more than 100 mg dm^−3^, in the top 25 cm layer of soil [15,16]. The concentration of aqua-coordinated Cd^2+^ in soil solution and the diffusion coefficient of Cd^2+^ in aqueous solutions regulate Cd^2+^ transport to the roots of plants [17].

Cd enters the roots through both symplastic and apoplastic pathways, in free ionic and chelated forms [18]. Cd generally accumulates in the roots [19]. The cell wall acts as the primary barrier to Cd mobility; thus, it is important in the adsorption of Cd in the roots. Therefore, Cd accumulation in the cell wall can reach 95% of the total Cd content of the dry material [20,21,22]. Heavy metal transporters generally show a broad substrate specificity, thus, the movement of Cd in the roots is proposed to be mediated by multiple metal transporters [23,24]. Among those, the ZRT, IRT-like protein (ZIP) family transporters play an essential role in translocating divalent cations across the plasma membrane [25,26,27]. ZIP transporters, such as barley (*Hordeum vulgare*) iron-regulated transporter (IRT) 1 and rice (*Orysa sativa*) zinc-regulated transporters (ZRT) 1, 3 and 4, were reported to facilitate the accumulation and transportation of Fe^+2^, Mn^+2^, Zn^+2^, Cd^+2^ and Ni^+2^ [28,29,30]. The ZIP family transporters OsNRAMP1 and 3 were also associated with Cd transport [31,32]. Beyond the ZIP family transporters, heavy metal-associated domain (HMA) family P-type ATPases are also important in Cd uptake and translocation in the roots of grasses. In rice, OsHMA1 is a tonoplast protein that regulates the release of Cd from the vacuoles of the root cells and, thus, controls the entry of Cd into the xylem. OsHMA3 is suggested to regulate Cd translocation towards the developing shoot tissues. Nevertheless, the mutations of OSHMA4 and 5 also suggest their contribution to Cd translocation [33]. The metal tolerance family protein *Oryza sativa* zinc transporter 1 (OZT1) also contributes to vacuolar Cd transport [34]. Although, yellow stripe-like (YSL) proteins are also important in metal–nicotianamine complex transport; their importance in Cd transport is hardly characterized. Nevertheless, citrate may facilitate the xylem transport of Cd [6,35].

Cd toxicity causes macroscopic symptoms in plants such as leaf chlorosis, wilting, necrotic lesions, a reduction in biomass, and the inhibition of root elongation [36,37,38]. In high doses, Cd primarily affects the root system by obstructing growth and impacting morphology [39]. Exposure to Cd resulted in the reduction of seed germination and a significant decrease in the length of the root and shoot in *Triticum aestivum* and in *Zea mays* [40,41,42].

In the shoot, Cd also impacts photosynthesis. The impairment in gas exchange and water stress induction are also outcomes of Cd toxicity in plants [39,43,44,45]. Although, Cd is a stable divalent cation under biological conditions; the disrupted metal homeostasis and perturbed metabolic functions lead to the generation of reactive oxygen species (ROS) [46]. As a result of lipid peroxidation, the accumulation of reactive aldehydes such as malondialdehyde (MDA) occurs [47]. Moreover, nitrate reductase activity is inhibited due to reduced nitrate absorption and transportation to the shoots [48]. Cd toxicity results in the imbalance of macronutrient and micronutrient uptake [49,50,51]. Cd was shown to interfere with the process of the uptake, transport and utilization of several elements such as calcium, magnesium, phosphorus, potassium and, especially, iron [5,52]. In dicot models, Cd affects the translocation of iron in an exclusive way [53], resulting in an induced iron deficiency in the shoot [54,55]. In the monocot model barley, Cd exposure also induces chlorosis in the lack of sufficient iron [56].

Multiple physiological mechanisms are involved in the Cd tolerance of plants, such as chelation in the cytosol and sequestration in the vacuole [18], the biosynthesis of high-affinity Cd-binding ligands, including carbonic and amino acids [57], glutathione and phytochelatins [58], and metallothioneins [59,60]. The sequestration of Cd chelates also aid in the restriction of Cd transport in plants [24,61]. Vacuolar compartmentalization was found to play a major role in Cd hyperaccumulation and hypertolerance in *Thlaspi caerulescens* and *Nicotiana tabacum* [62,63].

Phytoremediation is a plant-based technique of soil decontamination that uses metal tolerant plant species capable of extracting metals from the contaminated soil and accumulating them in the tissues [64,65]. High biomass-producing, fast-growing and perennial plants that were designed genetically or selected and bred for the production of energy may present a cost-effective and feasible solution in terms of phytoremediation [66]. Szarvasi-1 (*Elymus elongatus* subspecies *ponticus* (Podp.) Dorn; syn. *Thinopyrum obtusiflorum* (DC.) Banfi cv. Szarvasi-1) is an energy grass cultivar that prefers to grow on alkaline and sandy soil. It is well adapted to slightly salty areas, the severe conditions of drought, frost and flood, as well as possessing a high yield of biomass [67]. Szarvasi-1 was assessed for its natural ability to tolerate or accumulate different types of heavy metals such as Cu, Cd, Pb, Ni, and Zn from a nutrient solution, and it was found to tolerate well 0.01 mM Zn, Ni and Pb, but it was sensitive to Cu and Cd [68]. 

The priming of crop plants has become one of the most cost-effective strategies for improving stress tolerance. Moreover, in plants that are highly sensitive to extreme environmental conditions, the concept of priming at the developmental stages results in the enhancement of plant production without any kind of genetic manipulation [46,69]. Different substances act as stressors and bioregulators and play a vital role in accelerating the defense mechanism of plants [70]. Some of the priming agents can help the induction of several transcriptional modifications, epigenetic and post-translational protein modifications, which attribute long-lasting protection during the time of stress exposure. The downregulation of reactive oxygen/nitrogen/sulfur species generation and the upregulation of the expression levels of dehydrin and aquaporin were also conferred to some of the priming agents that are utilized to counteract the various types of hyperosmotic stress [71,72].

S-methylmethionine (SMM) is a universal metabolite that is produced from methionine in flowering plants. Though, much is not known about the physiological functions of SMM, but it is generally believed that it acts as a methylation agent and is involved in regulating the levels of S-adenosylmethionine (SAM) in plant cells [73,74]. SMM leads to the downregulation of monoamine oxidase activity and enhances levels of RNA, which signifies its potential influence on gene expression. SMM can be easily translocated by the xylem and phloem sap of plants, and its maximum amount was detected in young leaves. SMM acts as a precursor of dimethyl sulfoniopropionate, which is a potent osmoregulator during cold and drought stress [74]. Several studies have already demonstrated that when SMM is applied in micromolar or millimolar concentrations to the nutrient solution, it improves plant tolerance against mosaic virus infection, hypoxia and cold stress in some plant species [75,76].

All these previous studies lead to the idea that the exogenous supply of SMM might prove beneficial in promoting the growth and development of plants under different types of environmental stress conditions, including heavy metal stress. We have applied SMM for the priming of Szarvasi-1 energy grass prior to Cd exposure and hypothesized that the reduction in Cd stress effects might improve the plant’s performance under phytoremediation conditions. We measured the various growth and physiological parameters as well as the Cd content and the effect on nutrient balance of plants and proved for the first time that SMM might be a valuable tool to aid plant tolerance under heavy metal stress.

## 2. Results

### 2.1. Effect of S-Methylmetionine on the Plants

To test the effect of SMM on Szarvasi-1, we applied sole SMM treatment in the nutrient solution of the plants for 24 h. After one week of growth, the pH of the nutrient solution slightly increased for the SMM-treated plants compared to the untreated control (Ctrl), but the different SMM concentrations did not modify further the pH (Appendix A). Chlorophyll (Chl) content measured as SPAD index was statistically the same for all treatments, a slight increase at the 0.02 mM SMM concentration was noted (Appendix A). The maximal quantum efficiency of photosystem II (PSII) was practically the same for the Ctrl and all treatments and within the normal, unstressed range (0.79–0.81) (Appendix A).

### 2.2. Physiological Responses of Szarvasi-1 to Cd Treatment and Priming

Precultivated, one-month-old Szarvasi-1 plants were subjected to SMM pre-treatments for one day and then treated with 0.01 mM Cd, as described in Section 4.1. After two weeks, visible symptoms were observed on the treated plants. The Cd-treated plants without priming were smaller, and the leaves were pale-green or yellowish, as compared to the untreated Ctrl and those primed with SMM. The average number of tillers increased to 4.25 in the Ctrl, whereas in the Cd-treated ones, it remained at 3 (Appendix A). SMM pre-treatment, at all concentrations, restored the development of the plants that produced 4 tillers in the pots. The pH of the nutrient solution of the Ctrl plants was in the slightly alkaline range, but the Cd treatment and priming, with increasing concentration of SMM, resulted in monotonously decreasing values to neutral pH (Table 1).

The dry weight of energy grass was reduced significantly for roots and shoots in the plants treated with Cd, as compared to the Ctrl (Figure 1A). However, priming with SMM tendentiously, but not significantly, increased the dry weight of roots and shoots. The highest effect was recorded at 0.02 mM SMM concentration, resulting in a 50% larger shoot dry weight compared to Cd treatment without SMM priming.

Cd treatment resulted in a significant, more than 60% reduction in the stomatal conductance of the plants, as compared to Ctrl (Figure 1B), and the priming with three different SMM concentrations did not stimulate the transpiration, as compared to the Cd-treated plants. Despite marked changes in stomatal conductance, the RWC was unchanged at all SMM concentrations, a nonsignificant decrease of 4.3% was observed in plants only exposed to Cd (Figure 1C).

The total Chl concentration of the youngest fully developed energy grass leaves was reduced significantly in the plants exposed to Cd stress, as compared to the Ctrl (Figure 2). Pre-treatment with 0.02 and 0.05 mM SMM concentrations induced a significant, almost 50% increment in the Chl concentration. In parallel, pre-treatment with 0.1 mM SMM resulted in almost a complete offset of Cd stress with respect to the Chl concentration that remained almost similar to the Ctrl plants.

Under Cd stress conditions, plants encountered a significant reduction in photochemical reflectance index values (PRI), as compared to the Ctrl plants (Figure 3A). The reduced PRI values were moderately enhanced in the plants provided with SMM pre-treatment, suggesting that SMM helped in the alleviation of Cd stress at all three SMM concentrations and the values obtained are statistically significant.

Furthermore, the maximal quantum efficiency of PSII (F_v_/F_m_) was also influenced by Cd stress, and a slight statistically significant reduction was observed in plants exposed to Cd, as compared to the Ctrl (Figure 3B). SMM priming proved effective in achieving the complete restoration of the maximal quantum efficiency of PSII. 

The concentration of the lipid peroxidation by-product MDA was not modified significantly in any treatments, although in the plants exposed to Cd, a somewhat increased MDA content was observed, as compared to the Ctrl (Figure 3C).

### 2.3. Changes in the Ionomic Patterns

#### 2.3.1. Cd Accumulation and Its Effect on Element Distributions

The control plants had very low Cd concentrations in both roots and shoots, originating from the trace contamination in chemicals (Figure 4). The two-week Cd exposure to one-month-old plants resulted in 4.7 mg g^−1^ FW Cd accumulation in the roots and 115 µg g^−1^ FW in the shoot. Pre-treatment with 0.02 and 0.05 mM SMM significantly lowered shoot Cd concentrations by 45 and 46%, respectively. A total of 0.1 mM SMM also caused a considerable 22% decrease, but it was not statistically significant. The translocation factor (TF) and bioconcentration factor (BCF) also changed significantly under SMM pre-treatment (Table 2). In the plants pre-treated with 0.02 and 0.05 mM SMM, the TF dropped to 58% in both cases compared to plants treated only with Cd but raised again to 76% under 0.1 mM SMM priming. The BCF decreased similarly to 60 and 58% compared to Cd-treated plants after 0.02 and 0.05 mM SMM priming, respectively, but increased to 89% for 0.1 mM SMM.

Concerning the effect of Cd treatment on the nutrient uptake of the plants, we found a significant decrease in the concentration of K, Mg, Mn, Zn and P in the roots compared to the Ctrl (Appendix A). SMM priming at all application doses further decreased only the Zn concentration. The Fe concentration changed in the opposite direction. It increased considerably in the roots upon Cd treatment compared to the Ctrl. SMM treatments at all application doses further significantly increased it in the roots. In the shoots, we found that K, S, Cu and Zn concentrations were significantly decreased by Cd compared to the Ctrl, even after SMM priming with all concentrations. In the case of P and Mo, the decrease became significant only in the plants primed with SMM. Nevertheless, K, S and Mo continuously increased with increased SMM concentrations applied for priming. The Fe concentration in the shoots decreased under Cd treatment compared to the Ctrl, but it was not statistically significant. The application of SMM for priming resulted in a tendentious increase from the Cd-stressed level to that above the Ctrl’s Cd concentration, but these changes became significant only above a 0.02 mM SMM concentration.

#### 2.3.2. A Multivariate Data (Cluster) Analysis

To understand the mineral uptake pattern in a plant grown in different treatments, the clustering of 12 essential minerals, based on dissimilarity with one another, was carried out. From the clustering patterns (Appendix A), it was observed that there was a consistency in the mineral uptake pattern across different treatments. In total, there were 4 major clads in case of the shoot. Ca and P were grouped in the first cluster. The majority of the minerals such as Cd, Zn, Na, Mn, Cu, Mo and Fe were clustered in the second clad. Mg and S were clustered in the third clad while K was placed alone with maximum distance from the other minerals in the fourth clad. In the second cluster, the sub clustering pattern varied across different treatments; however, the dissimilarity between them was very low and, hence, insignificant.

In comparison, the root sample clustering gave different results. Although the majority of minerals such as Cu, Zn, Na, Mn, Mg, Ca and K were placed in the same clad/position as they were in shoot sample clusters, while position of minerals such as Fe, Cd, S and P changed with the change in the treatment. Fe was placed within the same clad as Cd in Cd, 0.02SMM+Cd and 0.05SMM+Cd-treated root samples, indicating similarity in their accumulation pattern. Fe in the 0.1SMM+Cd-treated root samples was placed in a separate clad with higher dissimilarity from other minerals when compared to other treatment clusters, indicating higher Fe accumulation in root samples with the increase in the SMM concentration. The S position in the clusters was inconsistent in root samples throughout various treatments. P was placed along with the Cd in all the treatments, except in the Ctrl.

#### 2.3.3. Principal Component Analysis

With the purpose of multivariate data exploration, based on variance analysis, principal component analysis (PCA) was used to understand the correlation between the Ctrl and Cd-treated plant’s mineral uptake, which is illustrated in Figure 5. For plotting the PCA graph, data were standardized by using standard Euclidian biplots. From the analysis of the scree plot, Axis 1 and Axis 2, having 43% and 23% resolution scores in the shoot graph and 58% and 21% resolution scores in the root graph, respectively, were chosen to plot the correlation coordinates. Considering the present data sets (Ctrl and Cd-treated plants), in shoot PCA, it was observed that Fe and S are negatively correlated with Cd, indicating that the higher presence of Cd in the plants reduces the accumulation of Fe in the shoot. Whereas, in root samples, Fe showed a positive correlation with Cd, indicating a higher Fe accumulation in Cd-treated roots compared to the Ctrl group. Additionally, the Ctrl plants are negatively correlated with Cd-treated plants and are closer to essential minerals in both graphs including Fe. Cd-stressed plants lean towards Cd, indicating its higher presence. Similarly, Cd-stressed plants are negatively correlated with Fe and other essential minerals such as K, Cu, Zn, Mn, and P (and the non-essential Na). Interestingly, while all the other minerals are negatively correlated with Cd, Ca is closer to Cd.

#### 2.3.4. Canonical Analysis of Variance

Furthermore, for investigating the effect of SMM priming on Cd-stressed plants, canonical analysis of variance (CAV) was performed on the mineral uptake data obtained from the applied treatments. Unlike PCA, CAV analysis provides information regarding correlation among different groups by grouping similarly treated plants together. For plotting the CAV graph, data were scored using spherized scores. From the scree plot analysis, Axis 1 and Axis 2, having 72% and 19% in the shoot graph, and Axis 1 and Axis 2, with 89% and 9% of resolution scores in the root graph, respectively, were chosen to plot the correlation coordinates, as illustrated in Figure 6. It was observed from both the shoot and root samples that the Ctrl, Cd-treated and SMM+Cd-treated groups are in three different corners, suggesting clear divergence among the groups. Similar to PCA plots, in CAV plots, the Ctrl plants are negatively correlated with Cd and positively correlated with essential minerals such as P, S, Mn, Cu and Zn. Cd-treated plants are positively correlated with Cd and Ca in shoot samples and with Cd, Ca and S in root samples. Cd-treated plants are negatively correlated with other essential minerals such as K, Cu, P, Mn, Zn and Fe (and the non-essential Na).

SMM priming, along with Cd treatment, brought plant groups closer to the origin, as compared to the Ctrl and Cd-treated groups in both shoot and root samples. The SMM+Cd-treated plant groups overlay on each other, indicating similarity between the groups. It also indicates that the difference in the concentration of SMM did not hold much effect on the mineral uptake in general. SMM treatment to plants increased the accumulation of Fe in both the shoots and roots and Na in the plants and, hence, is positively correlated with it closer in the graph. SMM+Cd-treated groups are also close to Cd-treated ones, indicating Cd uptake is a major influencing factor despite having SMM pretreatment. Furthermore, Mo in the shoots and Zn, Mg, Mn and K in the roots are strongly negatively correlated with SMM-treated groups. Mg shows a similar but less strong negative correlation in the shoot.

## 3. Discussion

### 3.1. Physiological Responses of Szarvasi-1 to Cd Treatment and Priming with SMM

Cd stress severely affects and leads to a potential impact on physiological and biochemical disorders in plants [5]. In the case of sensitive agricultural plants, the key consideration is what kind of defence procedures should be potentially used against plant stressors in order to protect the crops from severe damage. Biologically active compounds such as SMM and salicylic acid, or mixed compounds such as methyl jasmonates are known to have the potential to minimize damages in plants exposed to different types of abiotic stress [76,77,78]. In phytoremediation or recultivation procedures, the establishment of plantations and the survival or vitality of plants may depend on the harmful effect of heavy metals in the soil, which may be alleviated by various biologically active materials. In our present work, we hypothesized that exogenously applied SMM may have a potential role in the alleviation of Cd stress in Szarvasi-1 energy grass.

Prior to Cd treatment studies, the sole application of SMM was evaluated and we found that none of the measured parameters (pH in the nutrient solution, Chl content and the maximal quantum efficiency of PSII) showed significant deviation from the Ctrl (Appendix A). Thus, we have concluded that SMM can be applied as a priming agent in our hydroponic system without the risk of SMM-caused damages in the plants. Priming agents generally affect in a concentration-dependent manner [79,80]. The best dose could be different depending on the species, the application method or the actual environmental circumstances. SMM, like many other priming agents that can be considered as weak stressors, could be easily overdosed, especially in plants under stress; however, it is poorly described in the literature. The dose-dependent application of salicylic acid, a priming agent with many similar effects, is much more documented. Higher doses of exogenous salicylates can trigger oxidative stress, and therefore, they are generally applied in a concentration below 0.5 mM, and lower concentrations could be favourable for the stressed plants [81]. Lower concentrations of SMM also proved to be more beneficial for the Cd-challenged plants in the recent study, pointing at the dynamically dose-dependent manner of priming agents.

The toxic effects of Cd on Szarvasi-1 are in good agreement with previous findings [68,82,83,84]. Heavy metal stress in general hampers the antioxidative defence system via the overproduction of oxidants causing an oxidative burst in plant cells. Membrane lipid peroxidation and corresponding MDA accumulation is a good indicator of this process [85,86]. However, MDA concentration in the leaves of Szarvasi-1 showed only a slight, but not significant, increase under Cd stress. This may be explained by the effective antioxidative defence in these plants and that Cd treatment was applied at a one-month-old stage of development. Similar resistance has also been found in the biomass plants *Arundo donax* [87].

The pH change in the nutrient solution demonstrated that, in all treatments, the plants preserved the ability to maintain normal metabolic activity and raise pH to the slightly alkaline level to neutral range preferred by Szarvasi-1 [67,88]. Cd treatment and SMM priming followed a pattern of slight decrease in pH, probably referring to the metabolic shift, due to increasing SMM concentrations. This finding is in agreement with previous studies performed with wheat crop [89].

SMM priming helped to reduce changes in the multiple physiological parameters that were affected by Cd. A tendentious increase in the dry weight of roots and shoots (Figure 1A) and significantly higher Chl concentrations in the youngest fully developed leaves were measured at 0.02 and 0.05 mM SMM concentration with a complete retention of Chl at 0.1 mM SMM concentration (Figure 2). These findings accord with the previous results of Rácz et al. [90] and Páldi et al. [76], who demonstrated that SMM increases the stability of biomembranes in the chloroplast and throughout the cell and so helps in the maintenance of Chl content. The maintenance of the Chl concentration appropriate to the environmental conditions is important, as Chl-based light absorption generates the energetic driving force and carbon chains for metabolic acclimation to stresses. RWC was slightly decreased by Cd, but the Ctrl values were perfectly retained when SMM was applied (Figure 1C). However, stomatal conductance was found to be unaffected by SMM priming (Figure 1B), which may refer to the fact that SMM did not reach, directly, the autonomous guard cells, or the metabolic shift caused by SMM was not in relation with their functioning. Our results are in line with similar studies performed under NaCl stress with *Brassica napus* and *Cucumis sativus* plants, in which SMM had no stimulatory effect on stomatal conductance and transpiration rate [91,92].

Several parameters indicating plant stress were found to be alleviated by SMM priming, independent from the concentration applied (Figure 3). Photosynthetic light-use efficiency, which is inversely connected with carotenoid level in the leaves, indicated by PRI, doubled compared to the Cd-stressed level, but the Ctrl level was not totally retained. It was also evidenced in previous findings that SMM priming in wheat resulted in improved photosynthetic parameters [78]. Furthermore, the maximal quantum efficiency of PSII and the MDA concentration were completely retained by SMM pre-treatment in our study. These findings are in agreement with previous studies conducted by Páldi et al. [76] and Kósa et al. [93], in which SMM pre-treated plants had a higher value of F_v_/F_m_ as compared to plants subjected to cold stress, and by Fodorpataki et al. [91], in which SMM pre-treatment resulted in the decreased membrane lipid peroxidation of salt stressed plants.

SMM pre-treatment also mediates the lowering of electrolyte leakage of leaves after cold stress, which shows the positive effect of SMM pre-treatment on the stability of membranes [90,94]. Our findings are also in accordance with previous studies conducted on polyamine (synthesised via SMM cycle) treatment to prevent Chl loss and to preserve thylakoid membranes [95]. All these previous studies and our findings in the present work support the fact that increased Chl concentration in SMM-primed plants is at least partly due to the membrane-protective nature of SMM that contributes greater stability to thylakoid membranes.

### 3.2. Ionomic Background of the Induced Tolerance to Cd

Although, Cd lacks a specific transporter, it is transported by NRAMPs, IRTs, ZIPs, etc.; the substrates of which are essential minerals such as Zn, Fe and Mn, primarily [84]. This may lead to Cd accumulation in the roots and shoots, which was confirmed in the present study. SMM priming did not modify the Cd accumulation in the roots but significantly decreased its translocation to the shoot suggesting either an increased sequestration of Cd in the root cells or the enhancement of the transport of competitive essential minerals. Based on the multivariate analysis, Cd treatment increased S uptake in the root, which may refer to an increased thiol metabolism [96]. However, the S uptake was not significantly decreased by 0.02 mM and 0.05 mM SMM pretreatment or at all by 0.1 mM SMM pretreatment. This result does not suggest an SMM-related increase in the sequestration of Cd in the root cells. 

The correlation between Cd and other nutrient elements observed in the multivariate analysis refers to an altered uptake and translocation pattern. The Fe concentration was increased in the roots and decreased in the shoot by Cd, which may refer to an inhibited xylem loading of Fe. It has been reported that Cd causes reduced citrate transporter *Ferric Reductase Defective3* (FRD3) expression in the root xylem parenchyma of *Solanum torvum*, resulting in a decreased translocation of Fe [97]. In rice, the root pericycle cells citrate transporter ferric reductase defective1-like (OsFRDL1) are also specifically required for Fe translocation [98]. Due to this disturbance in the Fe homeostasis, the Cd-treated Szarvasi-1 plants became Fe deficient, which was indicated by low Chl concentration in the leaves. This finding is in good accordance with previous studies [54,55,68]. Furthermore, the uptake and accumulation of other essential minerals was significantly modified, which may be due to the direct effect of Cd on proteins or the indirect effect through the altered Fe homeostasis [84,99].

The effects of SMM priming on Cd-stressed plants can be clearly visualized on the CAV plots. The positive correlation between Fe and the SMM-treated plant groups infers that the pre-treatment with SMM improves the accumulation of iron in the shoot and, particularly, in the root. SMM is a non-proteinogenic amino acid known to play a role in plant metabolism by regulating genes/proteins [75,100]. SAM is the precursor of mugineic acids; the so-called phytosiderophores that are produced solely in graminaceous plants. These compounds are released from the roots to the rhizosphere to bind Fe and then reabsorb the complex (strategy II in Fe uptake) [101]. SMM and SAM interconversions have also been described [102]. We suggest that SMM may upregulate the phytosiderophore metabolism through a yet unknown metabolic pathway, through SAM leading to increased Fe accumulation in the roots of Szarvasi-1. This is supported by the increasing Fe level with the increasing SMM concentration applied.

Along with Fe, other nutrients have been affected by SMM priming, too. The SMM-induced increase in Fe uptake and translocation may be responsible for the decrease in Mn levels due to competitive transport on the same transport systems. From the CAV biplot, it was also observed that SMM treatment negatively affected the uptake of Mg and Mo in shoots and Zn in roots, which are also essential minerals in plant metabolism. Further research is required to establish a conclusive relation between these minerals and SMM treatment.

## 4. Materials and Methods

### 4.1. Plant Material and Treatments

For germination, uniform-sized healthy Szarvasi-1 seeds were selected and sterilized in 3% (W/V) sodium hypochlorite solution for 15 min and washed with deionized water two times. Seeds were germinated for seven days on wet filter papers in Petri dishes at room temperature and in sunlight. Three seedlings with root length of about 2–5 cm were put on a 2 cm wide rubber sponge strip, rolled up and fastened in a hole (35 mm in diameter) that was cut in a polystyrene plate. The polystyrene plates with 9 groups of seedlings were placed in plastic buckets that were filled up with 5 dm^3^ quarter-strength-modified Hoagland nutrient solution of the following composition: 1.25 mM KNO_3_; 1.25 mM Ca(NO_3_)_2_; 0.5 mM MgSO_4_; 0.25 mM KH_2_PO_4_, 11.6 µM H_3_BO_3_; 4.5 µM MnCl_2_ × 4 H_2_O; 0.19 µM ZnSO_4_ × 7 H_2_O; 0.12 µM Na_2_MoO_4_ × 2 H_2_O; 0.08 µM CuSO_4_ × 5 H_2_O and 25 µM Fe(III)-citrate-hydrate (basic nutrient solution). After 2 weeks, the volume of the nutrient solution in each bucket was increased to 10 L. The nutrient solution was continuously aerated and replaced with fresh solution once in a week. The plants were grown in a climate-controlled growth chamber at 20–25 °C, at 70% relative humidity and 150 µmol m^−2^ s^−1^ PPFD with 10–14 h dark/light period.

After 4 weeks of pre-cultivation, the group of 3 plantlets that were still wrapped together in the rubber sponge was transferred to polystyrene rings of approximately 8 cm in diameter and placed into 800 mL pots with same nutrient solution. Plants were treated with 0, 0.02, 0.05 and 0.1 mM SMM concentration for 24 h (priming). SMM used in the experiments was a synthetic, racemic mixture of D and L optical isomers (Merck, Darmstadt, Germany; Sigma-Aldrich cat. No. 64382). After 24 h SMM priming treatment, the nutrient solution that had SMM was removed, and plants, both primed and not primed, were exposed to 0.01 mM CdNO_3_ dissolved in Hoagland nutrient solution for two weeks preserving one set of pots without priming and Cd treatment as untreated Ctrl. Therefore, five experimental variants were set up during the experiment: the Ctrl provided only with Hoagland solution, a series exposed to only Cd, and three series exposed to Cd and primed with 0.02, 0.05 and 0.1 mM SMM (Ctrl, Cd, 0.02SMM+Cd, 0.05SMM+Cd and 0.1SMM+Cd, respectively). 

A preceding experiment was also conducted to check any possible effect of SMM on the plants. After 4 weeks of pregrowth, 24 h SMM priming treatments in 0, 0.02, 0.05 and 0.1 mM concentration were applied, but after the replacement of the SMM-containing nutrient solution, Cd was not added. Plants were harvested after 1 week.

### 4.2. Determination of Relative Water Content

For the measurement of *RWC*, plant samples were collected from the first fully developed leaves. Leaf material of about 20 mg was used. The *FW* of the leaf samples was determined, and leaves were incubated on a wet filter paper that was put in a Petri dish to reach full hydration in 4 h. After *TW* of the leaves was recorded, leaf tissues were put in an incubator overnight at 85 °C to attain a constant *DW*. The relative water content was calculated as follows:$$RWC\;\left(\%\right)=\frac{\left(FW-DW\right)}{\left(TW-DW\right)}\times 100\;$$

### 4.3. Determination of Chlorophyll Content

For the measurement of Chl concentration, plant samples were collected from the first youngest fully developed leaves. Leaf material of about 25 mg was used. Plant samples were homogenized in 80% (V/V) acetone. After homogenization, the samples were subjected to centrifugation for 5 min at 10,000× *g*, Chl concentration of the leaves was determined by using spectrophotometer (UV-2101PC, Shimadzu, Japan) according to Porra et al. [103]. Chl concentration was calculated on the FW. In the preceding experiment, Chl content of the first fully developed leaves were assessed by a handheld SPAD meter (Konika-Minolta SPAD 502+).

### 4.4. Measurement of Malondialdehyde Content

The estimation of the degree of lipid peroxidation was carried out by measuring the concentration of its by-product MDA. The assay was conducted according to Heath and Packer [104] with slight modifications. Leaf material of about 100 mg was homogenized at 4 °C in 1.25 mL of 0.1% (m/V) trichloroacetic acid (TCA) and subjected to centrifugation at 15,000× *g* for 15 min. After centrifuging the plant samples, 1 mL of 20% (m/V) TCA and 1% (m/V) thiobarbituric acid (TBA) were added to 250 µL of the supernatant. (Organic solvent-free aqueous solution of 2-thiobarbituric acid was prepared by directly dissolving the crystalline solid TBA in the aqueous system, immediately before use). At 90 °C, the solution was incubated in a water bath for approximately 1 h. Later, it was put on ice to terminate the reaction and to enable the spectrophotometric measurement. Absorbance was recorded spectrophotometrically using UV-2101PC at 532 nm (ℇ = 155 mM^−1^ cm^−1^). The concentration of MDA was determined on the basis of FW.

### 4.5. Determination of Stomatal Conductance

An AP4 porometer (DELTA-T Devices, Cambridge, UK) was used to measure the stomatal conductance on the adaxial side of the youngest, fully developed leaves. Stomatal conductance was calculated as mmol H_2_O m^−2^ s^−1^.

### 4.6. Chlorophyll a Fluorescence Induction

Chl *a* fluorescence induction measurement was conducted with intact leaves using a PAM 101-102-103 chlorophyll fluorometer (Walz, Effeltrich, Germany). Plant leaves were first adapted for a 20 min dark period. Determination of *F*_0_ level of fluorescence was carried out by turning on the measuring light (modulation frequency of 1.6 kHz and PPFD less than 1 μmolm^−2^ s^−1^) after 3 s illuminations with far-red light for eliminating reduced electron carriers [105]. The maximum fluorescence yield of the dark-adapted stage *F_m_* was measured by exerting a 0.7 s pulse of white light (PPFD of 3500 μmolm^−2^ s^−1^; light source: KL 1500 electronic, Schott, Mainz, Germany). The maximal quantum efficiency of PSII reaction centers was determined after the leaves were adapted to the dark for 20 min and calculated as:$$\frac{F_{v}}{F_{m}}=\frac{F_{m}-F_{0}}{F_{m}}$$

### 4.7. Photochemical Reflectance Index

PRI measurements were made on first fully developed leaves using Plant Pen PRI210 (PSI Instruments, Drásov, Czech Republic). *PRI* is a stress index sensitive to changes in carotenoid pigments and indicates changes in photosynthetic light-use efficiency and the rate of CO_2_ uptake [106]. The index was defined as:$$PRI=\frac{\rho 530-\rho 590}{\rho 530+\rho 590}\;$$
where *ρ*530 and *ρ*590 indicates reflectance at 530 nm and 590 nm, respectively.

### 4.8. Determination of Element Concentration

Elemental content of shoots and roots was measured after the acidic digestion of the plant samples. Dried (2 days at 85 °C) plant samples were subjected to digestion in ccH_2_O_2_ for 1 h, then ccHNO_3_ for about 15 min at 60 °C and 45 min at 120 °C. The solution was thoroughly homogenized, and eventually filtered through MN 640 W filter paper (Macherey-Nagel, Düren, Germany). ICP-OES (inductively coupled plasma–optical emission spectrometer, Spectro Genesis, SPECTRO, Freital, Germany) was used for determining elemental content of the filtrate. A multielement standard for 33 elements was used for calibration (Loba Chemie Product code: I166N, Loba Chemie PVT, Mumbai, India).

### 4.9. Translocation Factor and Bioconcentration Factor

TF of Cd was defined as concentration of Cd in the shoot divided by the concentration of Cd in the root, whereas BCF of Cd was defined as concentration of Cd in the shoot divided by the concentration of Cd in the nutrient solution.

### 4.10. Statistical Treatment

A total of 35 plants, which had 5 experimental variants, were applied. Each experimental variant contained 7 biological parallels. Samples were taken individually for all the measurements, and repetition was carried out 3 times. Measurements and sampling were made randomly in 5–7 parallel on middle sections of the youngest fully developed leaves individually, except for the element composition, for which whole root and shoot material was collected. The data presented are mean and standard deviation. Statistical analysis was performed by using software package InStat 3.0 (GraphPad Software, San Diego, USA) and the data were evaluated by analysis of variance (ANOVA), followed post hoc by Tukey–Kramer multiple range test. Statistically different groups (*p* < 0.05) are indicated by lowercase and uppercase letters.

For multivariate data analysis, SYN-TAX 2000 for Windows software (SYN-TAX 2000, ELTE, Budapest, Hungary, http://podani.web.elte.hu/SYN2000.html, (accessed on 11 October 2022)) was used. Clusters were formed using ‘HierClus’ application for every experimental variant, separately, to understand the difference in the clusters. UPGMA method was used for clusters and data were standardized with the standard deviation method. ‘Ordin’ application was used for calculating PCA and CAV graphs. PCA was carried out with Ctrl and Cd-treated plant samples with Euclidian distance as a parameter. CAV was conducted on all the five experimental variants with Euclidian distance as the set parameter. 

## 5. Conclusions

In this study SMM was applied as a priming agent, prior to Cd treatment, in hydroponics. We found that it exerted a positive effect on Cd-stressed plants, which retained physiological performance and growth. In particular, the protective effect was remarkable for the Chl concentration, photosynthetic light-use efficiency and maximal quantum efficiency of PSII. This ameliorative effect of SMM was supposed to be based on, at least partly, the lower root-to-shoot Cd translocation by the upregulated Fe uptake and transport. 

Priming with a low concentration of SMM helps decrease Cd accumulation in the shoot and, thus, protects plants against stress provoked by Cd translocation so that the protective effect of SMM may be potentially utilized in phytostabilization procedures in Cd-contaminated land. In high concentrations, SMM also exerts a protective effect, thereby maintaining higher Chl accumulation, but does not reduce Cd uptake effectively. 

## Figures and Tables

**Figure 1 plants-11-02979-f001:**
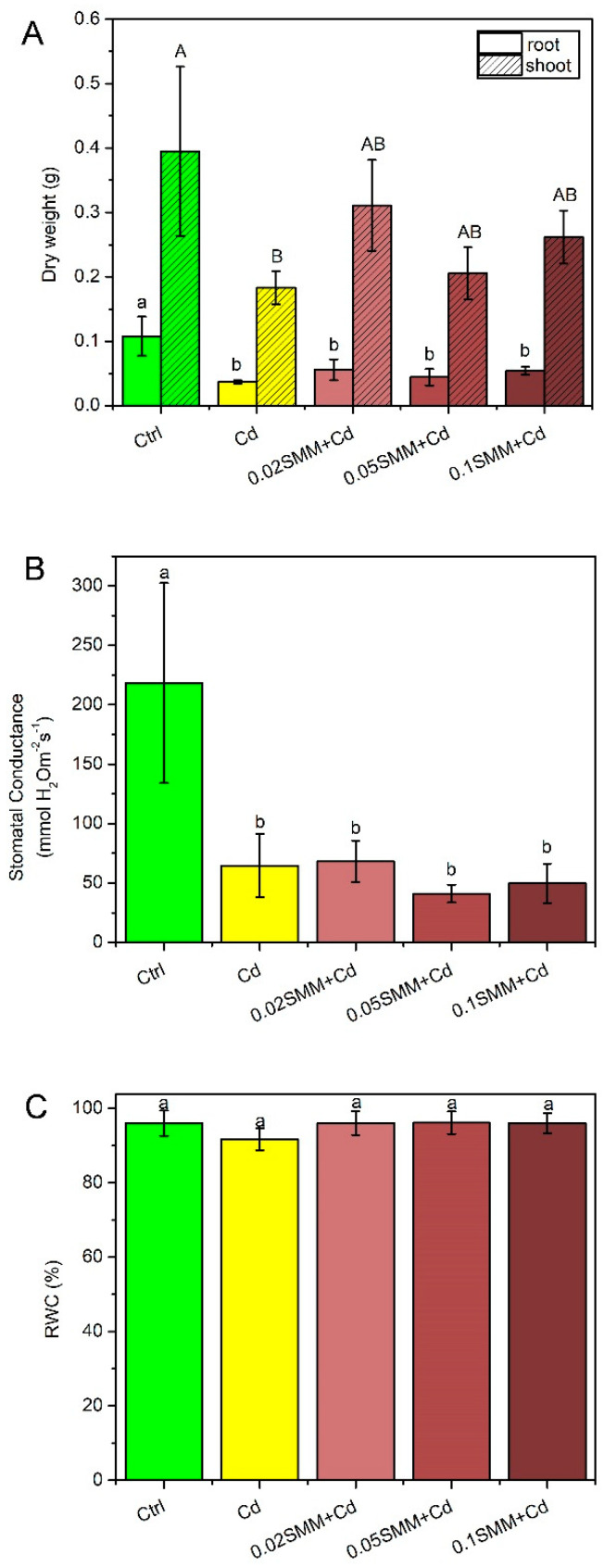
Root and shoot dry weight (**A**), stomatal conductance (**B**) and RWC (**C**) of the youngest fully developed leaves of Szarvasi-1 energy grass primed with 0.02, 0.05 and 0.1 mM SMM and exposed to 0.01 mM Cd. Error bars represent SD values. To compare the differences, one-way ANOVA was performed on each dataset combined with Tukey–Kramer post hoc test on the treatments (*p* < 0.05). Different lowercase and uppercase letters indicate significantly different groups.

**Figure 2 plants-11-02979-f002:**
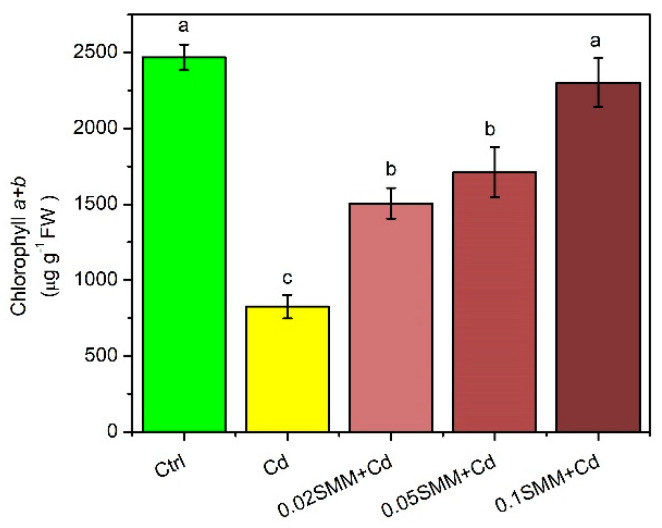
Total Chl concentration of the youngest fully developed leaves of Szarvasi-1 energy grass primed with 0.02, 0.05 and 0.1 mM SMM and exposed to 0.01 mM Cd expressed on a fresh weight (FW) basis. Error bars represent SD values. To compare the differences, one-way ANOVA was performed on each dataset combined with Tukey–Kramer post hoc test on the treatments (*p* < 0.05). Different lowercase letters indicate significantly different groups.

**Figure 3 plants-11-02979-f003:**
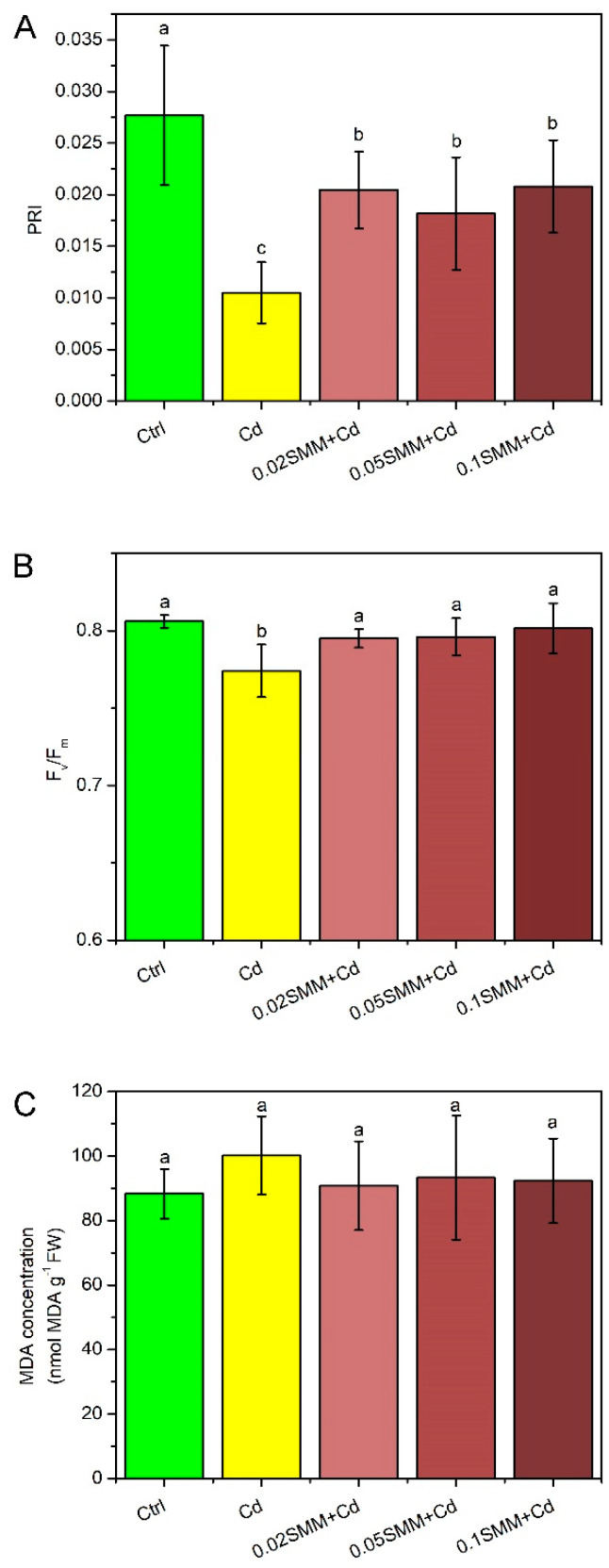
Photochemical reflectance index (PRI) (**A**), maximal quantum efficiency of PSII reaction centers (F_v_/F_m_) (**B**) and MDA concentration (**C**) in the youngest fully developed leaves of Szarvasi-1 energy grass primed with 0.02, 0.05 and 0.1 mM SMM and exposed to 0.01 mM Cd. To compare the differences, one-way ANOVA was performed on each dataset combined with Tukey–Kramer post hoc test on the treatments (*p* < 0.05). Different lowercase letters indicate significantly different groups.

**Figure 4 plants-11-02979-f004:**
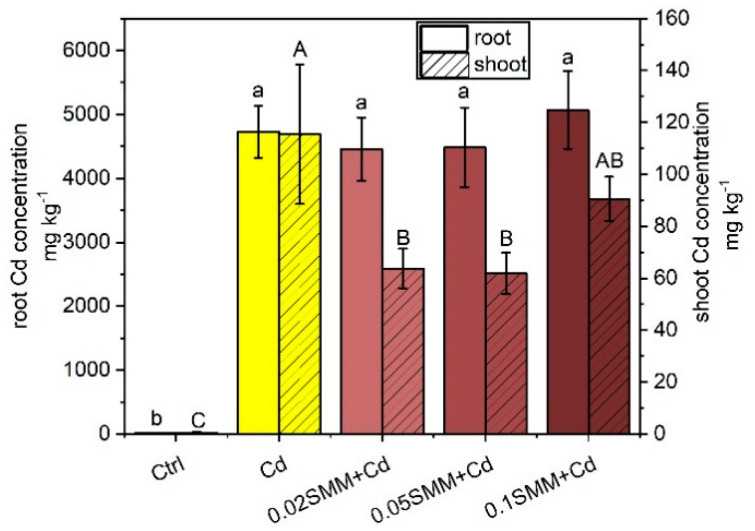
Cadmium concentration in the roots and shoots of Szarvasi-1 energy grass primed with 0.02, 0.05 and 0.1 mM SMM and exposed to 0.01 mM Cd. To compare the differences, one-way ANOVA was performed on each dataset combined with Tukey–Kramer post hoc test on the treatments (*p* < 0.05). Different lowercase and uppercase letters indicate significantly different groups.

**Figure 5 plants-11-02979-f005:**
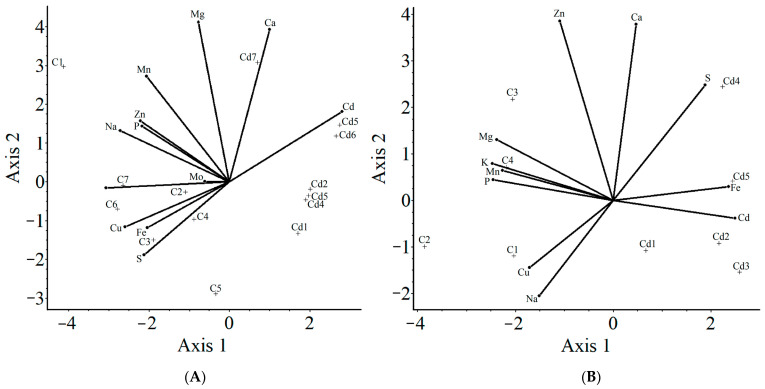
Principal component analysis of mineral uptake pattern in shoot (**A**) and root (**B**) samples of energy grass in Ctrl and Cd treatment. C refers to Ctrl.

**Figure 6 plants-11-02979-f006:**
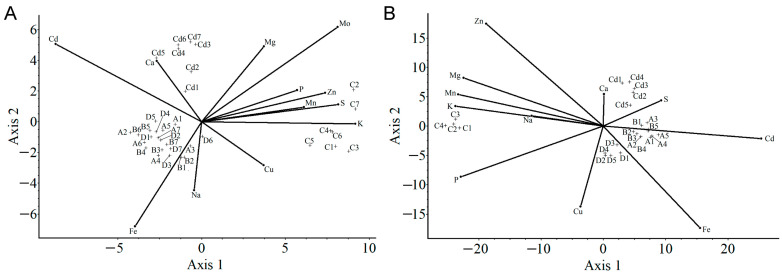
Canonical analysis of variance of mineral uptake pattern in shoot (**A**) and root (**B**) samples of Szarvasi-1 energy grass in different treatments. C, Cd, A, B, and D refer to Ctrl, Cd-treated, 0.02SMM+Cd, 0.05SMM+Cd and 0.1SMM+Cd groups.

**Table 1 plants-11-02979-t001:** pH values of the fresh bulk nutrient solution and after two weeks of 0.01 mM Cd treatment of Szarvasi-1 energy grass primed with 0.02, 0.05 and 0.1 mM SMM. Data are presented as mean ± SD. To compare the differences, one-way ANOVA was performed on the dataset with Tukey–Kramer post hoc test (*p* < 0.05). Different lowercase letters indicate significantly different groups.

Treatment	pH in the Fresh Solution	pH at Harvesting Day
Ctrl	5.90	7.25 ± 0.01 a
Cd	5.85	7.18 ± 0.03 b
0.02SMM+Cd	5.85	7.11 ± 0.02 c
0.05SMM+Cd	5.85	7.03 ± 0.01 d
0.1SMM+Cd	5.85	6.98 ± 0.02 d
0.02SMM	5.99	
0.05SMM	5.97	
0.1SMM	5.87	

**Table 2 plants-11-02979-t002:** Translocation factor (TF) and bioconcentration factor (BCF) of Cd in Szarvasi-1 energy grass primed with 0.02, 0.05 and 0.1 mM SMM and exposed to 0.01 mM Cd. To compare the differences, one-way ANOVA was performed on each dataset combined with Tukey’s post hoc test on the treatments (*p* < 0.05). Different lowercase letters indicate significantly different groups.

Treatment	TF	BCF
Ctrl	0.045 ± 0.037 *	
Cd	0.024 ± 0.006 a	102.66 ± 23.88 a
0.02SMM+Cd	0.014 ± 0.002 b	61.36 ± 13.76 b
0.05SMM+Cd	0.014 ± 0.001 b	59.72 ± 14.06 b
0.1SMM+Cd	0.018 ± 0.002 ab	91.52 ± 24.67 ab

* Ctrl was not involved in statistical treatment as Cd was only in trace contamination levels in the nutrient solution.

## Data Availability

The data presented in this study are available on reasonable request from the corresponding author.

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
