# Peer review of "S-Methylmethionine Effectively Alleviates Stress in Szarvasi-1 Energy Grass by Reducing Root-to-Shoot Cadmium Translocation"

_plants, 2022, doi:10.3390/plants11212979_

Round 1

Reviewer 1 Report

The aim of the paper is to demonstrate the effectiveness of priming with S-methylmethionine to counteract adverse effects of cadmium contamination on various physiological processes in a local strain of the energy grass species Elymus elongatus, which is suitable for phytoremediation of polluted soils. Even though cadmium toxicity for plants was thoroughly studied in the last decades, the main original contribution of the present work is the use of a naturally occurring amino acid derivative (SMM) in the priming process of plants to cope more effectively with heavy metal toxicity. The experimental design is appropriate to test the hypothesis. The figures properly show the experimental data. Conclusions are consistent with some of the evidence, but need to be improved. Discussion of some of the many experimental results should be also improved, as indicated below.

Specific comments:

- please clarify the Cd concentration used in the experiments, because in some parts of the manuscript (abstract, results: rows 173) it appears to be 0.1 mM, while in other parts (methods, title of some figures) it is indicated to be 10 µM (10 µM is not equal to 0.1 mM, but to 0.01 mM)

- row 65: “any” should be replaced with “and”

- row 79: please clarify briefly for the readers how can a tonoplast protein (and not a plasma membrane transporter) regulate the Cd entry into the non-living xylem vessels

- row 117: “grow in” should be replaced with “grow on”

- row 136: “involve” should be replaced with “is involved”

- row 139: “smoothly” should be probably replaced with “easily”

- row 153: in the given context “tolerance” seems to be a more proper term instead of “resistance”, because resistance implies avoidance of any harmful effect, while tolerance is achieved through acclimation to stress conditions (e. g. by metabolic modulation)

- row 184: please clarify how should one interpret the formulation “substantially but not significantly increased”, because usually an increment is substantial only when it is statistically significant

- rows 185-186: try to explain (in the section of discussions) why the lowest SMM concentration (0.02 mM) could cause the highest increase in shoot dry weight

- figures 1C and 3C might be omitted because of the low informative value (…nothing changes)

- row 200: “reduced” should be corrected to “was reduced”

- rows 222-225, 369-374, 402: if changes in MDA concentration caused by Cd were not statistically significant, then the statement that “priming with SMM restored the MDA values of the control” is not justified; if Cd did not lead to any significant change as compared with the control, it is not correct to state an alleviative influence of the priming agent (…because it is nothing to compensate for); please try to explain why such a high Cd concentration (0.1 mM or 0.01 mM for two weeks) did not cause any significant oxidative membrane damage (as reflected by the low MDA generation, rows 369-374), because this result is contradictory to many studies that have demonstrated that Cd leads to oxidative membrane damage (mainly peroxidation of unsaturated fatty acids) even in lower concentrations than the one used in the present experiment

- rows 232-234: please try to explain (in the section of discussions) why lower concentrations of SMM have a more pronounced lowering effect on shoot Cd content than higher SMM amounts (…the logic of this relationship is not obvious)

- row 264: “on the contrary” should be probably replaced with “on the opposite direction”

- row 264: why would SMM priming further decrease the Zn content – please try to find a plausible explanation in the section of discussions

- rows 282, 288, 312-313, 333: please document why Na was supplied to the nutrient medium of plants, and why Na is mentioned in the manuscript as an essential mineral nutrient (…”essential minerals like … Na”, rows 312-313), as sodium is not known to be an essential macro- or micronutrient for plants … this is why it is not present among the macro- and micronutrients of the Hoagland solution, being present at most as the sodium salt of molibdenate for the Mo ultramicronutrient

- please try to explain why in the presence of Cd more Fe accumulates in roots, considering that Cd may compete with Fe for import into root cells

- rows 383-385: please explain briefly (in the section of discussions) what is the significance of changes in the chlorophyll content of leaves, because there is no general principle according to which a higher or a lower chlorophyll concentration would be more beneficial for plants under certain stress conditions (except for acclimation to different light regimes)

- row 411: the term “thylakoid membranes” seems more suitable than “membranes of thylakoids”

- row 475: it is not clear why was light intensity set to 150 µmol m-2 s-1, which is a very low, rather limiting photon flux density for the grass species used in the experiments

- row 480: please clarify if the SMM used in the experiments was a synthetic, racemic mixture of D and L optical isomers or it consisted exclusively of the biologically active and naturally occurring L- enantiomer of the amino acid derivative S-methylmethionine; it might be important because in the former case (mixture of D and L forms) the biologically active concentrations would be two times lower than the indicated 0.02, 0.05 and 0.10 mM

- in the section of methods: how big could be the middle section of the first fully developed leaf of a grass species, if the same leaf was used for several extractions and determinations, such as RWC, chlorophyll content, MDA concentration? Please indicate what quantities of the same leaf were used for the different measurements (the used leaf weight is indicated only for MDA)

- row 514: please indicate the solvent used for preparation of 1% (m/V) TBA solution

- row 516: “stopping the reaction” seems more suitable than “cooling the reaction”

- row 541: please indicate what does the letter “p” represent in the formula

- rows 574-575: the first sentence of “Conclusions” should be omitted because it has nothing to do with the real conclusions of the presented experiments

- rows 579-580: there is nothing in the presented experiment to demonstrate the direct utilization of SMM through SAM, so this conclusion is rather speculative and not supported by the own data

- rows 585-586: and ambiguous relationship is stated between chlorophyll content and biomass production, please clarify it; why would a higher chlorophyll accumulation lead to higher biomass, considering that under proper light conditions a higher biomass can be achieved with low chlorophyll contents of leaves

- it would be beneficial to the readers to completely rephrase the conclusions, because the main findings of the very many experimental data, which were presented in the results section, are not mentioned at all in the present form

- in the section where results are discussed, several publications concerning physiological effects of SMM under abiotic stress conditions on membrane lipid peroxidation, on photosynthetic pigment contents, on the quantum efficiency of PSII, on growth parameters etc. are omitted; it would be useful to mention some of these references related to interpretation of the presented results.

Reviewer 2 Report

This manuscript (Rana et al.) describes the role of S-methylmethionine (SMM) application in reducing soot-to-root translocation of Cd in Szarvasi-1 energy grass. The authors found that this effect correlates with upregulated Fe uptake and transport. In treated plants, K, Mg, Mn, Zn, and P decreased in roots and K, S, Cu, and Zn in the shoots.

This study is important to understand and propose possible priming strategies to plant stress responses and to understand how plants respond to Cd stress. The effects of S-methylmethionine on plant physiology are intriguing and it is great that this study was conducted. While the underlying mechanism is yet unknown, this study will stimulate future research into this interesting plant. The authors did a good job describing the results and discussing them.  Overall, the paper is well-written, and the conclusions agree with the data presented.

My only suggestion for this kind of study would be that, when possible, adding images of the untreated and treated plants (even when no phenotype is evident) could be informative for other research groups.

Reviewer 3 Report

The topic of the manuscript is interesting and the authors showed that SMM priming exerts a positive effect on Cd stressed Szar-vasi-1 plants retaining physiological performance and growth.

The general layout of the manuscript is good.

English is good but a general review would improve it further.

I have only one criticism : the number of References cited is high and it could be reduced by selecting the most recent ones.

Reviewer 4 Report

The subject of the manuscript is interesting and consistent with the scope of the Journal.

The abstract faithfully conveys the scope of investigations and conclusions drawn. The keywords correspond well to the scope of the research.

I think the paper needs some corrections:

1) add detailed information about analytical quality control (CRM, standards, etc.), especially for cadmium analyses,

2) why did you use p<0.05 for the statistical analysis of the obtained test results? (in laboratory tests, p<0.001 should be used, and in vegetative pot experiments p<0.01)

3) remove “empty” places from the paper,

4) format References section according to Instructions for Authors,

5) format all sections of the manuscript according to Instructions for Authors.

You must check your paper very exactly and correct all mistakes and complete lacking data of papers.
